# Position: Not All Explanations for Deep Learning Phenomena Are Equally Valuable

Alan Jeffares [1]   Mihaela van der Schaar [1]

## Abstract

Developing a better understanding of surprising or counterintuitive phenomena has constituted a significant portion of deep learning research in recent years. These include double descent, grokking, and the lottery ticket hypothesis – among many others. Works in this area often develop *ad hoc hypotheses* attempting to explain these observed phenomena on an isolated, case-by-case basis. This position paper asserts that, in many prominent cases, there is little evidence to suggest that these phenomena appear in real-world applications and these efforts may be inefficient in driving progress in the broader field. Consequently, we argue against viewing them as isolated puzzles that require bespoke resolutions or explanations. However, despite this, we suggest that deep learning phenomena *do* still offer research value by providing unique settings in which we can refine our *broad explanatory theories* of more general deep learning principles. This position is reinforced by analyzing the research outcomes of several prominent examples of these phenomena from the recent literature. We revisit the current norms in the research community in approaching these problems and propose practical recommendations for future research, aiming to ensure that progress on deep learning phenomena is well aligned with the ultimate pragmatic goal of progress in the broader field of deep learning.

## 1. Introduction

Deep learning phenomena – that is, surprising empirical observations that appear to disagree with our existing expectations of how neural networks function – have become a

[1]Department of Applied Mathematics and Theoretical Physics, University of Cambridge. Correspondence to: Alan Jeffares <aj659@cam.ac.uk>.

*Proceedings of the $42^{nd}$ International Conference on Machine Learning*, Vancouver, Canada. PMLR 267, 2025. Copyright 2025 by the author(s).

> **Position**
>
> This work argues that many prominent deep learning phenomena discussed in the research literature are not representative of challenges encountered in real-world applications of deep learning. Thus, not all efforts to *understand* these phenomena are equal in value – we should focus on using them to refine our *broad explanatory theories* of important aspects of deep learning rather than developing *narrow ad hoc hypotheses* to describe them in isolation. However, this perspective is not consistently reflected in current research practices within the community.

prominent topic in deep learning research. In recent years, there has been a remarkable interest and excitement around several of these phenomena within the research community (e.g. Belkin et al., 2019; Frankle & Carbin, 2019), which has even begun to extend into the public consciousness (e.g. Heaven, 2024; Ananthaswamy, 2024). The sustained research focus on these idiosyncratic phenomena is heavily reflected in publications at leading machine learning conferences and community efforts such as workshop sessions.

**What type of "deep learning phenomena" does this work address?** As this terminology could refer to a broad range of empirical observations, we will first specify the kind of phenomena we address throughout this work. We adopt the definition used in the workshop for "Identifying and Understanding Deep Learning Phenomena" at ICML 2019 which called for "interesting and unusual behavior observed in deep nets" that can be "isolated and analyzed". We further narrow our scope to what we call *edge cases* i.e. phenomena that challenge our intuitions without appearing prominently in practical deep learning applications. For example, the emergence of *grokking* – that is, delayed generalization, long after perfect training performance (q.v. Section 2) – *is* included in our discussion as it has captured the interest of many in the research community despite not being a practical concern in training frontier models. We focus on these edge case phenomena as, despite their prevalence in research (which we later detail), it is not obvious *how* or *why* they should be studied.

**Where might research on these phenomena go wrong?**
Given that these edge case phenomena are, by definition, of limited relevance to practical settings, their role in deep learning research more broadly is worth revisiting. In particular, practitioners might ask when developing models for real-world applications (e.g. large language models in production settings), *is this phenomenon something I should be concerned with?* Consequentially, researchers might ask *what is the value in studying this phenomenon?* In this work, we posit that there is little value in pursuing a *resolution-oriented* approach to research in this setting. In other words, if a phenomenon presented a practical challenge then we might wish to *resolve* it (i.e. develop methods to mitigate its negative effects and amplify its positive effects) which could be supported by efforts to *understand* it isolation (by this we refer to the development of narrow ad hoc hypotheses that don't necessarily generalize more broadly – see Section 3.2 for further details on this terminology). However, as these phenomena *do not* pose a significant practical challenge, it should not be assumed that this approach will yield valuable outcomes. Indeed, without broader implications "explaining" an inconsequential phenomenon may be no more useful than solving a puzzle. However, this view is not reflected in current research practices on deep learning phenomena.

**What is the role of these phenomena in deep learning research?** When deep learning phenomena do not require resolution, we might be tempted to reason that research efforts are better directed elsewhere. However, in this position piece, we argue that these phenomena can still provide a valuable test bed for refining our *broad explanatory theories*[1] which encapsulate our fundamental understanding of core aspects of deep learning. In particular, they can act as extreme or synthetic settings that can stress test or provide counterexamples to our current beliefs about important mechanisms in deep learning. We argue that this can be achieved by integrating two key changes into our research practices. ▶ **A more pragmatic approach:** Although estimating the downstream impact of research is challenging, it is not entirely random. We suggest that a more critical consideration of the potential practical utility when pursuing research on a phenomenon can indeed result in more useful outcomes. While competing explanations for a phenomenon can clearly be evaluated based on their accuracy (i.e. are they both *true*?), it can also be the case that they are equally accurate but unequal in their utility (i.e. does this explanation provide insights that enable us to improve performance in practice?) Thus, researchers should be *pragmatic* in how they select among the vast space of potential

research directions, asking questions such as *does this new theory provide generalizable intuition beyond the specific phenomenon in which it is developed?* ▶ **A more scientific approach:** Paired with this pragmatic approach, we also advocate for the application of a more rigorous scientific methodology. The natural sciences, for example, have a long history of applying scientific practices, attempting to falsify and update competing theories of the world. We believe that better adapting these practices (e.g. preregistration) and working more collectively (e.g. greater efforts at reconciliation among different explanations), will result in research that is less disjoint and uncovers more general knowledge with broader utility.

**Goals:** The objectives of this position paper are threefold. **(1)** It aims to spark a conversation within the community as to *why* we should study deep learning phenomena at all, suggesting that greater emphasis should be placed on considering the potential for utility in this area; **(2)** It provides a critical viewpoint on three specific examples of these phenomena – namely, *Double Descent*, *Grokking*, and *The Lottery Ticket Hypothesis* – by challenging just how emergent they are in practical settings, but highlighting their potential for broader *pragmatic contributions* to progress in the wider field; **(3)** It provides actionable recommendations on how individual researchers and the wider research community might refine their research practices to better utilize these problems and maximize their potential value.

## 2. A Selection of Popular Deep Learning Phenomena

In this section, we explore three specific examples from the literature of deep learning phenomena. These particular examples were selected as they are representative of the kind of edge case phenomena we address in this work and due to their recent popularity within the research community. For each example we provide a general background on the phenomenon, some discussion on its *practical irrelevance* (i.e. its limited direct appearance in real-world applications ☹), and highlight its *broader value* (i.e. examples of the approach we promote where the tangible value extends beyond the phenomenon itself ☺). There is a much larger body of existing work on these topics and this section is not intended as a complete survey of that literature.

**Significance:** Before proceeding, let us first comment on the current significance of deep learning phenomena in the research community. Quantitatively, the original papers introducing each of the three specific phenomena that we discuss in this section (Belkin et al., 2019; Power et al., 2022; Frankle & Carbin, 2019) have been collectively cited 7272 times at present[2] (all since 2019) and continue to re-

---

[1]This term refers to the type of theories that provide insight more generally beyond the narrow setting which they describe. In Section 3.2 we make this distinction between *narrow ad hoc hypotheses* and *broad explanatory theories* more explicit and provide concrete examples.

[2]As reported by Google Scholar on 5 June 2025.

ceive in the order of thousands of citations per year. While citations are not necessarily a measure of significance, they do highlight the intense interest they have generated across the field. This interest is further evidenced by the sustained engagement from the community at major conferences. This is highlighted, for example, by the recurrence of dedicated workshops at each of the major deep learning conferences that explicitly call for works on deep learning phenomena. These include SciForDL at NeurIPS 2024, MechInterp at ICML 2024, or ME-FoMo at ICLR 2024. It is further emphasized by explicit references to these particular phenomena within the main track accepted papers at these same conferences[3] (149 at Neurips 2024, 132 at ICML 2024, and 108 at ICLR 2024). With this context in place, we believe it is reasonable to assert that deep learning phenomena hold a significant position within the research community, and prompting a dialogue to challenge current practices in this area is both timely and valuable.

## 2.1. Double Descent

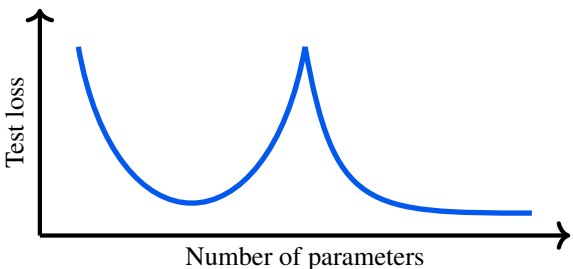

*Figure 1.* **Double descent.** As a neural network is re-trained with increasing parameter count it follows a traditional U-shaped curve in test loss followed by an unexpected second descent.

**Background:** A conventional understanding of statistical learning describes a compromise between underfitting and overfitting where a model has sufficient capacity to model the data-generating process but insufficient capacity to take the shortcut of blindly memorizing the training labels (e.g. Hastie & Tibshirani, 1990). This is often illustrated as a U-shaped curve in test error as we increase model capacity, as shown in Figure 1. *Double descent* is a phenomenon coined by Belkin et al. (2019) (although it had been noticed earlier; see e.g. Loog et al. (2020)) where it was observed that as the *parameter count* continues to grow, eventually test performance will again improve resulting in a distinctive *second descent*. This sparked considerable interest among researchers, as it appears to defy our traditional understanding of generalization and scaling.

---

[3]As measured by exact match for the phrases "double descent", "grokking", and "lottery ticket" within the text of the accepted papers at each of these conferences. We count any paper that matches at least one of the three phrases.

☹ **Practical Irrelevance:** A subtle but important caveat is that double descent occurs when model complexity is tracked using *parameter count*, which may not be a good measure of model complexity in non-linear models (Curth et al., 2024b). When appropriate regularization is applied, as is typically done both explicitly and implicitly through e.g. early stopping and stochastic gradient descent, the double descent effect is no longer expected to appear (Nakkiran et al., 2021; Hastie et al., 2022). This may explain why this double descent behavior generally *does not* appear in empirical scaling analyses of neural language models (Kaplan et al., 2020; Hoffmann et al., 2022) or vision transformers (Zhai et al., 2022).

☺ **Broader Value:** The appearance of double descent has directly motivated several methodological developments with broader implications. Examples include layer-wise learning rate heuristics (Heckel & Yilmaz, 2020), a new complexity measure that captures a divergence between train and test complexity (Jeffares et al., 2024a), and a framework for measuring the generalization gap (Nakkiran et al., 2020). Likely of even greater impact, this phenomenon has played a key role in a recent critical reevaluation of several aspects of our understanding of the core principles of deep learning in practice. These include the role of memorization in learning (Feldman, 2020), underspecification of a solution (D'Amour et al., 2022), benign overfitting (Bartlett et al., 2020), and the bias-variance tradeoff (Adlam & Pennington, 2020).

## 2.2. Grokking

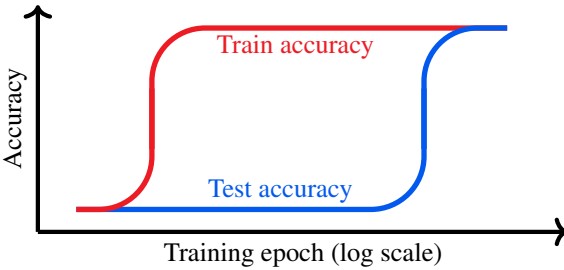

*Figure 2.* **Grokking.** A neural networks train set accuracy reaches 100% early in training, but a similar level of generalization performance measured on a test set only occurs much later.

**Background:** Grokking was introduced in Power et al. (2022) where it informally describes the general phenomenon of a model achieving "generalization far after overfitting". A prototypical example is provided in Figure 2 where a neural network obtains perfect training accuracy with poor test performance early in training, and much later obtains a corresponding sharp rise in test performance. This effect disagrees with typical large-scale deep learning intuitions, where test curves closely follow training curves, and suggests that the common practice of stopping early may

preclude the discovery of a better solution. However, this definition is not strict, and certain descriptive characteristics have evolved in the literature as being associated with grokking. The effect is typically considered more "grokking-like" when there is a larger time gap between strong train and test performance (Liu et al., 2022a), near-perfect train performance rather than stagnation at some arbitrary level (Charton, 2024), and a more abrupt jump in test performance (at least under a particular metric such as accuracy) (Nanda et al., 2023).

☺ **Practical Irrelevance:** The settings in which grokking can be shown to occur are typically limited to small algorithmic datasets. As the dataset grows, the grokking effect is generally found to be less extreme (Liu et al., 2022a). Attempts to observe standard grokking in realistic scenarios have so far been unsuccessful (e.g. Dziri et al., 2024). While grokking can technically be induced on other data modalities, this is synthetically produced using a greatly inflated initialization scale of the network parameters (Liu et al., 2022b) and any known studies *not* using this mechanism report standard generalization (i.e. no grokking; Varma et al., 2024). Even on algorithmic data, Miller et al. (2023) show that grokking may also be induced by certain *concealed* data encodings. Furthermore, Kumar et al. (2023) note that the striking visual nature of the effect can often be attributed to the choice of metric while standard loss plots are less remarkable – this echos a similar point which has been made more generally on the so-called *emergent abilities* of large language models (Schaeffer et al., 2024). Additionally, Pope (2023) assessed several notable grokking papers and raised some concern that "the extremely simplified domains in which grokking is often studied will lead to biased results that don't generalize to more realistic setups."

☺ **Broader Value:** Grokking suggests that if we observe surprising qualitative jumps during training, then we need to reevaluate and improve our internal performance progress measures to better capture the true, steady, underlying improvement (Nanda et al., 2023). This has inspired a more holistic evaluation of learning dynamics through several perspectives including *lazy to feature learning* (Kumar et al., 2023; Lyu et al., 2024), *effective parameters*, (Jeffares et al., 2024a), *slingshots in the parameters norm* (Thilak et al., 2024), and *representation learning* (Liu et al., 2022a). On the methodological front, grokking encouraged Prieto et al. (2025) to highlight numerical instabilities in the *Softmax* function and develop a more stable alternative. Beyond its relatively narrow original setting and extending upon the original idea of grokking, recent works have shown that certain desirable quantities can continue to improve after standard validation performance has plateaued – i.e. adversarial robustness (Prieto et al., 2025) and out-of-domain performance (Murty et al., 2023).

## 2.3. Lottery Ticket Hypothesis

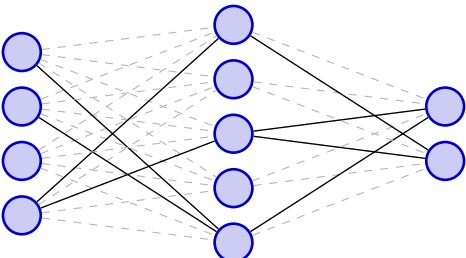

*Figure 3.* **Lottery ticket hypothesis.** A lottery ticket, where many network parameters have been pruned at initialization, will reach equal performance to its dense version after both are trained.

**Background:** The lottery ticket hypothesis was introduced by Frankle & Carbin (2019) and posited that in any dense randomly initialized neural network there exists a smaller subnetwork (or *lottery ticket*) that, if trained in isolation, will match the accuracy of the original network after training for at most the same number of iterations. If true, and if these subnetworks could be identified before training, this would suggest that neural networks could be trained more efficiently from the start rather than relying on post-hoc pruning of a dense model to find a sparse solution. Evidence from the original work, supported by subsequent empirical (e.g. Zhou et al., 2019) and theoretical (e.g. Malach et al., 2020) studies, strongly indicates that the phenomenon exists broadly and the hypothesis holds.

☺ **Practical Irrelevance:** Unfortunately, identifying these lottery tickets *before* training is highly non-trivial. Their existence was originally demonstrated by first training a dense network, applying standard pruning techniques *after* training, rewinding the network to its initialization, and then replicating the same pruning at initialization before retraining. This approach is highly sensitive to key training hyperparameters (Ma et al., 2021; Liu et al., 2019) and, although valuable for evaluating the hypothesis, is clearly not practical for real-world applications. Despite significant efforts to develop methods for discovering these subnetworks a priori, progress toward a viable technique has been limited and the lottery ticket approach has not materialized as a practical method (Frankle et al., 2021). Furthermore, it is unclear to what extent the potential gains in training efficiency due to sparsity could even be realized on modern hardware (Chen et al., 2022). As Jonathan Frankle, the original proposer of the hypothesis, summarized: "Today I really don't think people should be working on lottery tickets as a research area, I just don't think its the most interesting problem out there. But that said, I do think there are a lot of things that we should take away from that paper that should inform how we do science and what science we do going forward" (Frankle & Bashir, 2023).

☺ **Broader Value:** While directly identifying lottery tickets before training remains impractical, the underlying ideas have significantly influenced our explanatory theories of sparsity, pruning, and network efficiency. For instance, they have shaped our understanding of the relationship between sparsity and training dynamics, informing both post-training and during-training sparsification techniques (Hoefler et al., 2021). Beyond pruning, the core intuition that only a small subset of weights is critical for performance has been successfully integrated into alternative efficiency methods such as quantization (Lin et al., 2024) and parameter-efficient finetuning (Zaken et al., 2021). Additionally, lottery tickets have played a role in influencing impactful innovations in both vision (Touvron et al., 2021) and language models (Dao et al., 2022), where strategic weight selection and efficient computation led to practical improvements. A more comprehensive survey is provided by Liu et al. (2024), which highlights their influence on multiple research directions, even as directly using lottery tickets remains problematic.

## 3. On The Source of Value in Studying Deep Learning Phenomona

In the previous section, we highlighted that much of the practical value derived from studying deep learning phenomena has emerged as a byproduct of the research process, rather than from directly resolving these phenomena in practice. In this section, we build on this observation and argue for a more deliberate and systematic approach to studying these problems to maximize their value. However, before proceeding, we must first discuss how progress in deep learning is *defined* and how it can be *achieved*.

### 3.1. Sociotechnical Pragmatism and The Science of Deep Learning

The purpose of any scientific field is a challenging concept upon which this paper does not attempt to resolve or provide normative judgment. Instead, it is reasonable to suggest that Watson et al. (2024)'s characterization of *sociotechnical pragmatism* (i.e. the principle that research value is rooted in its downstream impact, versatilely encompassing both technical progress and societal considerations) is broadly compatible with the intuitions and incentives of much of the research field and reflects a flexible categorization of its values. Built upon certain intuitions from the tradition of pragmatism (Legg & Hookway, 2024), this framework is derived from the perspective that "conceptual advances are only valuable insomuch as they are useful. A theory with no practical implications is little more than a formal exercise" (Watson et al., 2024). In particular, the *value* of knowledge gained about a phenomenon is determined by its downstream utility – with the definition of utility sufficiently flexible to capture a broad spectrum of perspectives

of progress. This viewpoint is reflected in how we typically discuss progress in the field in terms of open problems and the pragmatic progress that is made on them (e.g. Kaddour et al., 2023). A recent work that systematically extracts and analyses the stated values in machine learning research papers finds utility-centric concepts to be the highest ranking of all values with *performance* and *generalization* appearing in 96% and 89% of papers respectively and *applies to real world* being explicitly stated as a value in over half of papers (Birhane et al., 2022). The perceived virtue of the pragmatic approach is reflected in several calls for its greater application in various areas of the field including causality (Loftus, 2024), network intrusion detection (Apruzzese et al., 2023), and medical applications (Starke et al., 2021).

The mechanism through which this sociotechnical pragmatic progress is made is a separate problem. However, it is clear that the principles of scientific inquiry, as previously discussed, are entirely compatible with this approach. The scientific method – when applied in the deep learning setting – can be understood[4] as a process that consists of refining an explanatory theory by iterating between (1) critical evaluation of said theory based on empirical (or other) evidence and (2) updating to an improved theory that better aligns with the existing evidence without diminishing its degree of falsifiability. The role of pragmatism in the process is through the choice of *which* theories are worth pursuing in the space of all possible theories, and which specific aspects of the theory are most valuable to critically evaluate and update – this is determined by an estimation of their expected utility. The significant role of an empirical scientific method in machine learning has been pointed out at least as far back as Langley (1988) and more recently has been discussed in terms of how it transfers to deep learning (Forde & Paganini, 2019), in terms of systemic proposals for expanding its recognition the field (Nakkiran & Belkin, 2022), and critiques of current practices (Herrmann et al., 2024; Karl et al., 2024; Gencoglu et al., 2019; Trosten, 2023). This is further cemented by community efforts such as the recent Workshop on Scientific Methods for Understanding Deep Learning at NeurIPS 2024.

### 3.2. Narrow Ad Hoc Hypotheses vs Broad Explanatory Theories

Even with careful application of scientific principles, not all explanations of a phenomenon carry equal utility. Popper

---

[4]More specifically, Pearson (1892) states "The scientific method is marked by the following features: (a) Careful and accurate classification of facts and observation of their correlation and sequence; (b) the discovery of scientific laws by aid of the creative imagination; (c) self-criticism and the final touchstone of equal validity for all normally constituted minds." While the philosophy of a universal definition might be challenged, this viewpoint is heavily reflected in the practice of modern science today (Hepburn & Andersen, 2021).

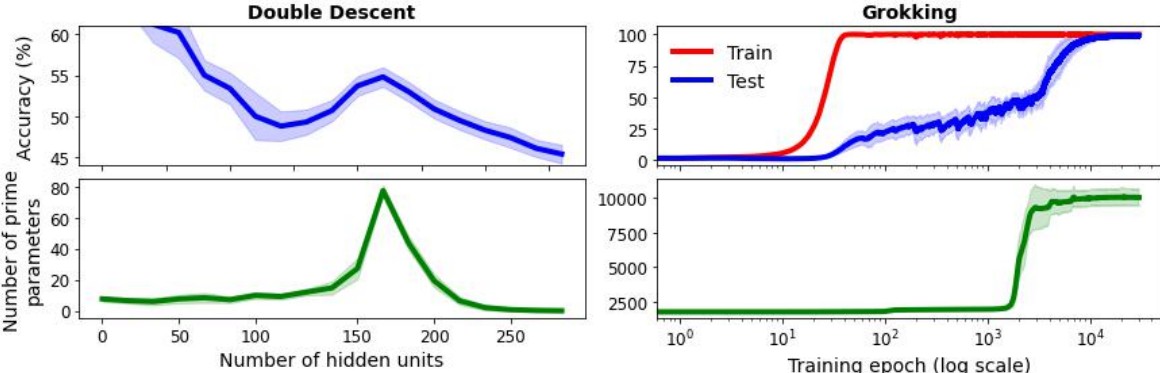

*Figure 4.* **A Unified Theory of Double Descent and Grokking Through *Prime Network Parameters*.** Examples of double descent (left) and grokking (right) from the recent literature are "explained" through the number of prime numbers in the network's parameters (after rounding; bottom row). In both cases, this metric accurately tracks test performance, capturing the effect of the phenomenon *without access to the test labels*. This hyperbolic theory illustrates exactly the type of *narrow ad hoc hypothesis* we advocate against. Although technically true and perhaps eye-catching, this approach relies on fitting a narrow theory to the observed phenomena post hoc, which is highly unlikely to carry broader implications that might lead to practical utility.

(1935) described a problematic approach to resolving challenges to our existing theories in which new hypotheses are, on the fly, *tacked on* in order to account for new observations that don't fit our existing understanding. He criticized these ad hoc patches as they have "no falsifiable consequences but merely served to restore the agreement between theory and experiment." These were later characterized as *ad hoc hypotheses* (Leplin, 1975) and only serve to "diminish the degree of falsifiability or testability of the system in question." To be pragmatic in our approach, we must distinguish between the differing expected utilities of potential theories.

Inspired by Popper's critique, in the context of deep learning phenomena, we refer to *narrow ad hoc hypotheses* as this style of custom, isolated explanations for unexpected empirical phenomena. These explanations are sufficiently overfit to the setting of the phenomena upon which they are developed that they cannot provide reliable implications to the broader field. They are scoped only to provide a shallow resolution. To illustrate this point, we develop our own example of one of these hypotheses in Figure 4. There, we show that both double descent and grokking can be "explained" as a consequence of the number of prime numbers in the network's parameters (after rounding). We might hypothesize that the optimization dynamics subtly favor prime-valued weights due to hidden arithmetic symmetries in the training data, and that phase transitions in test accuracy coincide with changes in the density of primes within the parameter vector. Although not strictly speaking *untrue* (i.e. prime weights *do* indeed track the effects), this somewhat absurd theory is unlikely to provide broader utility to the field. For example, naive methods derived from this theory might attempt to regularize the number of prime parameters in a neural network – a methodological dead end. There are countless theories such as this that can be used to describe a phenomenon after the fact, but their value cannot be assessed from how well they describe the data upon which they were developed – they *must* have predictive power more broadly.

Alternatively, we consider a different strategy of using empirical phenomena to advance our *broad explanatory theories*. This refers to our understanding of more general aspects of deep learning in which broader theories can be developed, updated, or falsified in response to a deep learning phenomenon. Rather than seeking niche explanations for an observed phenomenon, we wish to deepen our practical understanding of the more general underlying principles (e.g. *scaling* in double descent or *optimization* in grokking). For example, double descent gained interest as it appeared to disagree with the *bias-variance tradeoff* (Belkin et al., 2019), a fundamental principle which describes how models are expected to perform as they scale (or, equivalently, as the dataset scales). The bias-variance tradeoff implies a convex relationship between model complexity and predictive error, which has real implications for practitioners. That is, as we increase model complexity to the point where performance begins to deteriorate (i.e. overfitting begins), we should not expect further increases in model complexity to yield improved performance. If this principle is no longer true, then it may still be worth investing resources in attempting further increases in complexity, seeking that second descent in predictive error. Refining a broad explanatory theory (in this case, the bias-variance tradeoff) on the basis of the incongruent observations that emerge from a deep learning phenomenon (here, double descent) is exactly the approach for which we advocate in this work.

### 3.3. Towards More Intentional Research of Deep Learning Phenomena

With these foundations in place, we return to the main thesis. Our position is that, due to their lack of emergence in practical settings, research efforts on deep learning phenomena should critically assess their *value*. As a field that can be broadly characterized as following a philosophy of *sociotechnical pragmatism*, practical utility drives progress in deep learning. This is best achieved by approaching deep learning phenomena as a *scientific* endeavor in which we attempt to generate useful knowledge. Critically, this requires differentiating between *narrow ad hoc hypotheses* and *broad explanatory theories*. The former seeks shallow explanations that lack broader insights, while the latter refines a kind of universal understanding from which we can derive practical utility. Although determining an explanation's exact location on the spectrum between these two categories involves a degree of subjectivity, we claim that these efforts are essential. In this way, we believe that a more intentional approach to studying deep learning phenomena can greatly improve the collective research value attained in this area.

Although aspects of this position might appear somewhat abstract, in practice, its implementation can be relatively intuitive. For any empirical phenomenon of a neural network, we always have access to one particular explanation that perfectly explains the phenomenon – but only in the most narrow, ad hoc fashion. That is, the exact mathematical explanation on the level of inputs, outputs, parameters, activations, and gradients. Specifically, given the network parameters and any input, we can calculate the precise value of every activation as the input is passed forward through the network, leading to perfect knowledge of the output. Furthermore, for any output, we can calculate its corresponding loss and the resulting gradients, providing us with the exact expression for how every parameter will be updated in response. In other words, we have a complete mathematical expression for how a neural network behaves. Although trivial, in some sense, this is a perfect "explanation" of a neural network as it precisely predicts *any* empirical phenomenon – at least, at this level of detail. However, it is immediately obvious that this granular explanation is lacking in some way. We might ask ourselves, *what makes this explanation weak?* The answer is that, for many open problems in deep learning, this low-level description doesn't carry *utility*. It cannot teach us general principles from which we make practical advancements. It describes what we observe without providing a useful understanding. As with the prime network parameters example from Figure 4, we already have a natural intuition that some explanations are better than others. In this position paper, we sought to make this assessment more explicit while studying deep learning phenomena. We have encouraged a more deliberately prag-

matic approach to research in this area, where we consider the potential downstream value of our work.

Once we have integrated a sense of utility-driven prioritization of the directions we consider pursuing in our research, we must also follow some process in our research. As discussed in Section 3.1, for the process of developing, refining, and interrogating our broad explanatory theories, a scientific methodology is appropriate. Therefore, we should also strive to apply best practices for the scientific method, as we iterate between updating and critiquing our current theories. The general principles of conducting this process have been comprehensively discussed in the general scientific literature (e.g. Carey & Carey, 1994) and somewhat discussed in the context of deep learning (e.g. Forde & Paganini, 2019). In Section 5 we ground the discussion in this section by laying out some practical guidelines for pragmatic science in the specific case of deep learning phenomena.

## 4. Alternative Views

We now take some time to address a selection of reasonable points of objection to certain aspects of the position taken in the work.

**(a)** *The purpose of research is not necessarily pragmatic, some research is exclusively driven by intellectual curiosity.*

Exclusively curiosity-driven research is not uncommon in certain subfields of pure mathematics or philosophy, for example. However, although curiosity plays an integral role in the research process, it is less common to pursue it *exclusively* without any consideration of practical value in the deep learning context where the vast majority of researchers typically wish their work to have *some* downstream utility (even if that may come in the very long term). As discussed in Section 3.1 and supported by prior studies (e.g. Watson et al., 2024; Birhane et al., 2022), the norms and incentives within the field of deep learning are broadly aligned with pragmatic goals. We do not wish to dissuade researchers from pursuing research that aligns with their personal interests, we only encourage a greater consideration for potential practical impact within this research.

**(b)** *The position taken in this paper is not new, many researchers already approach deep learning phenomena aware that their utility lies in their broader implications.*

While we acknowledge that several individual works approach deep learning phenomena with an awareness of their broader implications, as highlighted in Section 2, this perspective is neither universal nor consistently reflected in research practices. Many studies focus on narrowly resolving individual phenomena without explicitly connecting their findings to broader theories or practical insights. While this claim is ultimately subjective, the lack of alignment between

the guidance in Section 5 – which outlines the practices that should be followed if this perspective is known – and typical papers on the topic suggests that, even if accepted, this perspective is not consistently reflected in published research.

**(c)** *Research is a speculative process, we cannot know which explanatory theories will be more useful a priori.*

While estimating the downstream impact of research is challenging, it is certainly not entirely random. For example, the structure of academic funding (both public and private), although imperfect, is typically predicated on our ability to focus research on areas that are most likely to provide practical impact on the particular areas of interest to the funding body. On an individual level, researchers are constantly choosing which research questions to pursue. As with other research outcomes such as "publication quality findings" or "a high citation count", we believe a researcher's ability to predict potential for "greater downstream utility" to be greater than chance and suggest that a *noisy* estimate of utility should not be mistaken for a *random* estimate. We hope that this position piece can offer an entry point to a dialogue on this topic among researchers.

# 5. Practical Considerations for Pragmatic Research of Phenomena

With our position on the current status of research on deep learning phenomena established, we now shift our focus to the future. We believe that, when approached with an awareness of their role within the broader field, these phenomena present an excellent opportunity for interesting and impactful research. In this section, we first highlight the distinctive characteristics of these problems that make them a compelling research focus in the age of large-scale deep learning (Section 5.1) and offer practical recommendations to maximize utility for future work in this area (Section 5.2).

## 5.1. Deep learning phenomena as a compelling research area in modern machine learning

● *Computational accessibility* - In modern deep learning research where neural networks have parameter counts in the order of billions, many aspects of research have become inaccessible to much of the research community due to prohibitive computational costs. Deep learning phenomena, by design, are extracted into their most simple setting and are typically approachable with even the most modest of resources. This provides an opportunity for meaningful contributions from even the most under-resourced in the community.

● *Low knowledge barrier for entry* - Much of deep learning research requires insider knowledge of the latest tips and tricks in terms of optimization strategies, architecture heuristics, benchmark choices, etc. Again, due to their pref-

erence for being studied in the simplest possible setting, deep learning phenomena tend to be primarily studied in a minimalist setting. Thus the barrier to entry is relatively low with the knowledge required to reproduce these canonical examples generally contained in standard textbooks (e.g. Prince, 2023).

● *Scientific inquiry over methodological competition* - Unlike many other areas of deep learning research, which often revolve around competing over state-of-the-art (SOTA) benchmarks, studying deep learning phenomena shifts the focus towards a scientific approach. This approach emphasizes collaborative knowledge creation over competitive SOTA chasing and can incentivize researchers to ask interesting questions and report findings rather than propose methods that presuppose a particular outcome. This potential to ask careful questions in which many outcomes are interesting reduces the pressure to achieve positive results which has been highlighted as a causal factor of poor reproducibility (Pineau et al., 2021).

● *Intersection of mathematical theory and empiricism* - While modern settings are often too complex to study using many forms of theory of a mathematical nature (e.g. Roberts et al., 2022), deep learning phenomena are typically studied in simplified settings that are more amenable to this type of analysis. This results in a problem that is interesting and approachable for both empirical and theoretical methodologies and provides an opportunity for cross-pollination and collaboration between the two. Several interesting works have already spawned at this intersection (e.g. Adlam & Pennington, 2020; Hastie et al., 2022).

## 5.2. Practical Recommendations for Impactful Research

In this section, we provide some practical recommendations for pragmatic research on deep learning phenomena built on the discussion up to this point – which we organize into three categories. We also provide a *self-evaluation checklist* in Table 1 that contains a selection of questions for researchers to consider throughout the research process.

**(1)** *Identification and Cataloging* - We begin by encouraging a greater emphasis to be placed on developing a complete descriptive exploration of a given phenomenon. This could include understanding the precise *setting* in which the phenomenon arises (e.g. data modalities, architectures, optimization approaches) and attempting to measure the factors that influence its *extent* and *fundamental characteristics*. It may be beneficial to deconstruct a phenomenon into its basic features and analyze how these elements are impacted by common considerations such as regularization or model size. Current practices often lead to new papers having to rediscover or reimplement phenomena in potentially disparate cases. Centralizing efforts to identify and catalog phenomena through open-sourcing high-quality and

extendable code implementations could reduce redundancy and individual errors, thereby improving research efficiency and fostering collaboration. Similar to how high-quality benchmark datasets have advanced methodologically focused research, enhancing the quality and range of shared canonical examples of deep learning phenomena could encourage more impactful work in this area. Subtle shifts in publication norms to better recognize the value of such contributions might help support these efforts (e.g. by explicitly rewarding systemic *exploratory studies* of this nature).

**(2)** *Prioritizing utility* - Reflecting the discussion in Section 3, we encourage researchers to consider the potential for downstream utility when studying and explaining a given phenomenon. It can be helpful for research to explicitly discuss the *purpose* of studying a phenomenon and how it may relate to broader goals such as model generalization, robustness, or interpretability – a practice that is typically treated as an afterthought, if included at all, rather than a central focus in recent works. These connections should play a meaningful role throughout phenomenon-based research and heavily influence how the paper is formulated, investigated, and executed to maximize its relevance and impact on the broader field. Researchers might find it valuable to prioritize extensions of existing ideas or novel ideas with generalizable implications over narrow explanations that apply only to the specific studied setting. Additionally, placing greater emphasis on relating existing theories more comprehensively can provide an environment where their relative strengths and weaknesses can be better exposed. At present, there is limited effort to compare and contrast different perspectives, which can make it challenging to critically assess them in relative terms. While measuring potential utility is inherently complex and often subjective, taking a thoughtful, case-by-case approach can still yield meaningful progress and is far more productive than neglecting the question altogether. While many of these suggested practices may appear rudimentary and broadly relevant, a survey of the literature suggests that they are not widely utilized in works that address deep learning phenomena – precisely where their implementation would be *most* impactful.

**(3)** *Following scientific principles* - As we are advocating for deep learning phenomena to be approached as a scientific exercise, adapting the best practices of that approach is essential. These best practices have already been heavily discussed in the scientific literature since at least Dewey (1910) and also in the machine learning literature (e.g. Forde & Paganini, 2019). At a practical level, they include principles such as hypothesis-driven research (Forde & Paganini, 2019), reporting of negative results (Karl et al., 2024), falsifiability (Gal, 2015), improved reproducibility (Pineau et al., 2021), preregistration (Hofman et al., 2023), replication and meta-studies (Herrmann et al., 2024). Additionally, they should avoid the pitfalls of commonly applied poor practices (Trosten, 2023). In the specific case of deep learning phenomena, we believe that certain aspects of these principles are particularly relevant. For instance, we should prefer theories that are more general and predictive beyond the setting in which they are designed but are also falsifiable (as the famous adage[5] goes "a theory that explains everything, explains nothing"). Great attention should be paid to evaluating the scope and setting of a phenomenon, rather than showing it only under contrived circumstances. Moreover, fostering a culture of collaboration over competition – such as through shared and unified benchmark-style repositories for code and data to reproduce and build upon existing work – can encourage a more collective effort towards pragmatic knowledge creation and reduce redundancy and fragmentation in isolated studies. Finally, we hope that this approach may provide an opportunity to draw inspiration from the rich history of other fields, particularly the natural sciences, in how they approach empirical phenomena and perhaps adapt aspects of certain successful strategies where applicable.

## 6. Conclusion

Deep learning phenomena occupy a growing corner of deep learning research, yet their collective characteristics and broader value are less often examined. In this paper, we have asked what purpose these phenomena serve and how they might meaningfully contribute to the wider goals of deep learning research. Although we have challenged the frequency at which several of these phenomena appear in practical settings, we argue that they can still play a significant role – supporting broader, pragmatic progress as objects of scientific inquiry. We hope this work encourages further reflection within the community on how we should best pursue research on deep learning phenomena in the future and where exactly their value lies.

**Broader relevance:** While this paper has focused on deep learning phenomena, we believe that key aspects of the perspective offered here can extend more broadly across deep learning. In many settings *beyond* edge case phenomena, periodically revisiting the fundamental purpose of research can lead to better practices and result in more valuable outcomes. For example, in the case of *emergent abilities in large language models* (Berti et al., 2025), distinguishing between genuinely concerning behaviors and contrived artifacts parallels the kind of practical relevance assessment discussed here. In such cases, a more consciously pragmatic approach may also help guide research priorities.

---

[5]This quote is often misattributed to Karl Popper, but in fact dates back at least as far as Society (1897) who attributes it to John Playfair in turn.

## Impact Statement

This paper explicitly focuses on increasing the practical impact of research on deep learning phenomena. However, the definition of impact used is broad and non-prescriptive. While utility could be defined in terms of improving predictive performance it could also be defined in terms of e.g. producing more equitable outcomes from a model. We do not take a position on this discussion in this work and believe these ethical and societal concerns should be carefully considered on a case-by-case basis.

## Acknowledgments

We wish to thank Paulius Rauba, Alicia Curth, Robert McHardy, Boris van Breugel, and the anonymous reviewers for their invaluable comments on earlier drafts of this paper. Alan gratefully acknowledges funding from the Cystic Fibrosis Trust.

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

## A. Additional Discussion

**Phenomena Representativeness:** The three phenomena we discuss (double descent, grokking, and the lottery ticket hypothesis) were chosen as they are widely recognized, popular, and have sparked significant discussion in the field. They also naturally fit into our specific focus as edge case phenomena i.e. those that challenge our intuitions without appearing prominently in practical deep learning applications. However, this list is not exhaustive, and we expect that there are numerous other examples that are relevant to this discussion – particularly those that have received less attention. We have aimed to ensure that the general content of the paper remains relevant for these other phenomena in addition to new ones that may emerge in the future.

**Non-edge Case Phenomena:** Although this paper has focused on edge case phenomena in which the observed effect doesn't appear widely in more practical deep learning settings, there are many unexpected phenomena that also arise outside of this setting (presumably many more, in fact). Unlike for edge cases, in phenomena more broadly, there may be significant value in resolving the phenomenon itself, even narrowly, in such a way that doesn't provide broader insight. However, we suggest that in many cases, these phenomena may still benefit from the approach described in this work. For example, the remarkably consistent effectiveness of deep ensembles (Lakshminarayanan et al., 2017) can be understood through several perspectives. Classic ensembles have been understood from the perspective of averaging out weak learners and increasing their representation capacity across the wide hypothesis space (Dietterich, 2000) or through a more expressive bias-variance tradeoff (Curth et al., 2024a). Recently, in the setting of *deep* ensembles of neural networks, additional explanations have been proposed that address important specific aspects of their success. For example, they can be understood as an emergent equivariance to symmetries in the data that do not hold for a single network (Gerken & Kessel, 2024) and, simultaneously, as an implicit optimization of a collective ensemble loss term (Jeffares et al., 2024b). Despite deep ensembles' success being prevalent in practical applications (and, thus, not an edge case), approaching these theories using the principles described through this work may help prioritize some over others in a useful way.

**Additional Note on Prime Numbers Example:** The double descent experiment uses the setting described in Greydanus & Kobak (2024) and the grokking experiment uses the setting of Power et al. (2022) based on the implementation in `https://github.com/teddykoker/grokking`. In our discussion, we dismissed the results of the prime numbers experiment in Figure 4 without further explanation of *why* prime numbers track these effects in this way. Although striking upon first viewing, in reality, these results are simply an artifact of the non-uniformity of prime numbers over an evolving distribution of network parameter magnitudes. In particular, neither 0 nor 1 is prime, so when the average magnitude is small, we expect to see fewer primes. In the grokking example, most parameters are initialized at each layer as $Unif(-\frac{1}{\sqrt{\text{num params}}}, \frac{1}{\sqrt{\text{num params}}})$, which will round to non-prime and only grow later in training, correlating with the increase in test accuracy. In double descent, there is also an interaction with the growing *number* of parameters since we are plotting the total number of primes. While the prime number explanation can therefore be dismissed as being due to a confounding variable (i.e. parameter magnitudes), it may also be the case that even explanations based on the confounding variable itself don't provide much broader insight, and thus also don't qualify as a useful explanation. We also note that results were averaged over 10 seeds, with a standard deviation displayed.

| Research Phase | Category | Self-Evaluation Questions |
|---|---|---|
| Early Stage Problem Identification | Context & Literature Review | • How have others framed or approached this phenomenon in the past?
• In which modalities, architectures, optimizers, etc. has it been observed thus far?
• What are both the most *simple* and *complex* settings in which this phenomenon can be observed?
• How has this phenomenon been understood in the past and what are the limitations of these explanations? |
| | Research Value Assessment | • Does this phenomenon appear in practical, real-world settings or is it an *edge case*? Where does its *value* lie?
• If it is an edge case, how synthetic are the conditions required to induce this phenomenon?
• What theories or assumptions does this phenomenon challenge either implicitly or explicitly?
• Optimistically, what are some of the greatest impact outcomes that could be achieved by making progress on this phenomenon? |
| Throughout the Research Process | Continuous Reassessment of Utility | • What predictions does my explanation make beyond this specific setup?
• Is practical utility being considered in selecting which research questions are being pursued?
• Have I controlled for confounding factors like metric choice, initialization scale, or overfitting?
• Are there measurable sub-patterns or intermediate behaviors worth isolating?
• How is this analysis evolving? Does it continue to hold promise for contributing to broader explanatory theories or has the scope drifted? |
| | Scientific Methodology | • What hypothesis (if any) is being tested — and is it falsifiable?
• Am I remaining conscious of common cognitive biases such as confirmation bias or false causality?
• Am I following other important scientific principles as described in Section 5.2? |
| Concluding Considerations | Cataloging & Reconciliation with Literature | • Are my findings reproducible, and is my code (or data) easily reusable and extendable?
• Did I document negative or ambiguous results that may be valuable?
• Have I explicitly reconciled my findings with previous works? Where do they agree and disagree?
• Have I been clear about which claims are empirical, theoretical, or speculative? |
| | Avenues to Practical Utility | • Have I explicitly stated just how these findings may contribute to broader explanatory theories? How are these findings *useful*?
• If utility is not immediate, have I articulated a vision for future relevance?
• Have I helped clarify *when* and *why* this phenomenon matters (or doesn't)?
• How transferable are these findings across e.g. architectures, tasks, or training regimes?
• Concretely, what promising future research may this work contribute to? Can I map out a speculative path to those contributions? |

*Table 1.* **Self-Evaluation Checklist.** A curated selection of key questions for researchers to consider at each stage of studying deep learning phenomena. Although there may not be strictly "correct" answers, considering these points may help align research with the position of this work.

