# OpenReview forum: "Position: Not All Explanations for Deep Learning Phenomena Are Equally Valuable"
_ICML.cc/2025/Position_Paper_Track — ICML 2025 Position Paper Track oral_

### Official Review · Reviewer_MHTA · 2025-03-13

**Significance:** 2
**Argument Clarity:** 3
**Rating:** 2
**Confidence:** 3

**Questions:**

How do the authors see the connection between their position and the opposition between reductionism and complexity science?

Can the authors reduce their hedging on the importance of all research being practical vs broad research for its own sake? Or if not willing to reduce the hedging reduce the moments where the paper acts as if pragmatism is the only worthy work in Deep Learning?

**Discussion Potential:**

2

**Paper Summary:**

The paper defends that the study of Deep Learning Phenomena, roughly speaking counter-intuitive or surprising properties of Deep Neural Networks, should focus on phenomena with practical significance and should aim to explain more generally a set of phenomena, instead of narrowly focusing on specific cases.

## update after rebuttal
I believe the paper is unclearly written and sadly cannot accept it as it stands

**Position:**

Yes

**Position In Title:**

Yes

**Related Work:**

3

**Strengths And Weaknesses:**

The paper is well-situated with the literature, focusing on 3 specific surprising properties of DNNs and the research that has spawned around it. The paper also grounds itself in interesting arguments about what science and research are for and how they should be conducted

On the other hand the paper seems to push too strongly the position that science needs to be done with a focus on pragmatism, which is not common in many areas related to Deep Learning, like Mathematics, Computer Science, and Physics. At the same time there are quite a few moments where there's somewhat excessive hedging, which leads one to be confused about the position of the authors, for example when they say "While working on deep learning phenomena we should prioritize the potential for downstream utility of our work (even if that is tangential and long term)", where by adding both "tangential" and "long term" to their recommendation we end up with an extremely broad if not empty recommendation.

Finally the authors push heavily towards the creation of general theories and considers that position as uncommon because it doesn't seem to be done, but there's no discussion on the possibility that it's not done out of the complexity of dealing with many phenomena at the same time, nor does it dialogue with literature comparing reductionism to complexity science, which seems to be at the heart of the contrast between "zooming in" on a single problem and trying to understand it, with the hope of eventually composing the theories into one unified explanation, versus trying to understand multiple phenomena all at once;

**Support:**

2

---

> ### Author Rebuttal · Authors · 2025-04-01
>
> We thank this reviewer for their helpful comments and clear suggestions for improving the paper.
>
> **W1 - Pushing for science to be done with a focus on pragmatism**
>
> Let us clarify our position on this point. We argue that deep learning is largely a utility-driven field which is quantitively supported by previous analysis of stated values (Birhane et al., 2022). We also argue that these stated goals are currently not being effectively attained in the deep learning phenomena literature (Sec 2). We then propose that changes in research practices could better obtain these outcomes. Thus, our pushing for pragmatism is only saying that *as this is a stated goal of the field, then we ought to have a dialogue about how well it is actually being realized.*
>
> However, we agree that some small minority of ML research is truly conducted without any consideration of potential utility (not to be mistaken with speculative research which pursues ambitious projects with a small probability of large impact). Our position paper is more relevant to the majority that is interested in considering practical utility and how it is best achieved. We are also non-prescriptive in the choice of what *type* of utility may be pursued as highlighted in our impact statement.
>
> We will update the text (particularly AV(a) on L323) to ensure this is clear.
>
> **W2 - Hedging of position**
>
> We thank the reviewer for highlighting that they had this impression. While we wish to be careful and nuanced in our claims, we did not intend to overly hedge our position. In the highlighted sentence we wished to clarify that more speculative research targeting "moonshot" projects may also be considered pragmatic in expectation (also mentioned in our previous answer) as we were concerned a reader could mistakenly believe we were arguing against long-term research. We will update this and other points in the text that may be perceived as hedging. If the reviewer has other specific examples they wish to point out we would be very grateful.
>
> **W3 - General theories and reductionism vs complexity science**
>
> When we refer to these broader *explanatory theories*, we do not require grand theories of all of deep learning and agree that such a reduction would not be a reasonable ask. Rather, we primarily refer to predictive theories of how certain important aspects of deep learning are functioning (although mentioned in footnote 1, we recognize its fundamental importance and will add a more detailed exposition in the text).
>
> For example, the bias-variance tradeoff provides an explanatory theory with broad, established relevance for overfitting in machine learning. Double descent appeared to challenge a conventional understanding of this theory and called for either an update to the theory or some resolution. As a result, our general understanding of the bias-variance tradeoff has improved, with downstream consequences for practical applications. This is precisely what we are advocating for and provides an excellent motivating example (see [1] for a concrete case).
>
> In contrast, suppose we ran a double descent experiment and found that, at the peak of the curve, the (rounded) network parameters contained more prime numbers. Although this might be a cute correlation, developing an explanatory theory from this is unlikely to be fruitful. We would not expect resulting methods (e.g. regularization methods for fewer prime parameters) to be generally effective and, thus, its utility is limited. As we do not wish to target individual papers to obtain an example of low-utility research, [we have invented a novel "theory" as a concrete illustration at this link](https://imgur.com/a/riZpNph). Although clearly exaggerated, this example illustrates the range of potential utility across conceivable explanatory theories.
>
> We appreciate the reference to the literature on reductionism vs complexity in science. Relating to philosophical works (by e.g. Thomas Nagel) and applied discussion on the topic in other fields would add some broader context to this aspect of the paper. In light of our previous discussion, we note that our work doesn't require any strong reductionist claims beyond the idea that we can better understand specific aspects of deep learning through certain evolving explanatory theories -- which should be uncontroversial as it is effectively an axiom of the field. We will integrate the connection to this literature into the main text.
>
> Motivated by the helpful feedback received in these reviews, we will add concrete examples to help ground our discussion and further improve the clarity of our arguments.
>
> **Questions**
>
> Regarding the specific questions asked, we hope this discussion provides a satisfactory answer and appreciate the suggestions which have improved the clarity and contextualization of the work. We would be happy to address any remaining concerns.
>
> [1] https://arxiv.org/abs/2011.03321

---

### Official Review · Reviewer_X44g · 2025-03-15

**Significance:** 4
**Argument Clarity:** 3
**Rating:** 4
**Confidence:** 3

**Questions:**

One reason that theorists focus on these things is *because they can be analyzed* using standard tools and techniques. When I work with mathematicians on the theory of ML, they ask "can we prove anything here?" And they are less interested if my answer is "probably not." Is there a way to construct or find theoretical problems in ML that are important in the sense of this paper and also subject to analysis and proof?

**Discussion Potential:**

4

**Paper Summary:**

This paper argues that the theory of ML is focused on particular technical points that are off-topic for performance and for improving ML; it argues for re-prioritizing theoretical questions. The paper particularly highlights three examples.

## update after rebuttal
My recommendation to accept remains. The paper is improved by the interactions with the referees. And I re-emphasize that this paper will generate good discussion at the meeting.

**Position:**

Yes

**Position In Title:**

Yes

**Related Work:**

3

**Strengths And Weaknesses:**

Strengths:

It is undeniably true that the theory of ML is only approaching very clean parts of very clean problems, and it hasn't delivered much of technical (or "clinical" one could say) value to the ML community.

The three highlighted phenomena do seem to be somewhat over-discussed, and somewhat irrelevant to the practice of ML.

This position will produce tons of discussion at coffee and dinner.

Weaknesses:

It's not obvious to this reader how much the three points of theory that have been highlighted are representative. That said, I don't have a way of generating more "statistics" on theory directions.

I don't like the authors use of "scientific." I think most places they use this they mean "engineering" or "clinical." The difference between science and engineering, imho, is that engineering is about making things work better, and science is about understanding things that can be understood. When physics switched from Newtonian gravity to GR, everything worked *way worse*. GR was a new understanding, not a way to make things work better (try calculating anything about the Solar System in GR; yes I am an expert on this). Also, science is interested in things like white dwarf interiors, which will never, in any way, impact humanity. That is, science isn't motivated by use value. So I think the paper is arguing for a more clinical approach, not a more scientific approach. Focusing on double descent is *exactly* what a scientist would do (and indeed, many do).

**Support:**

2

---

> ### Author Rebuttal · Authors · 2025-04-01
>
> We sincerely appreciate this reviewer's positive feedback and the thoughtful points raised in their review. We are particularly glad to hear their exceptionally positive view on our work's potential to produce interesting discussion.
>
> **W1 - Clarification on Representativeness of the Three Highlighted Phenomena**
>
> We understand the point being made regarding the representativeness of the three phenomena we discuss (double descent, grokking, and the lottery ticket hypothesis). We chose these examples because they are widely recognized, popular, and have sparked significant discussion in the field. However, we do not claim that they represent the full spectrum of deep learning phenomena. Our goal was to use them as illustrative cases that highlight a broader trend in theoretical research. These phenomena are not exhaustive of the field, but they serve to highlight our concerns. We have aimed to ensure that the general content of the paper remains relevant for other existing phenomena in addition to new phenomena that may emerge in the future.
>
> **W2 - "Scientific" vs. "Clinical" Approach**
>
> We appreciate the point raised with our use of the term "scientific" and the distinction between science and engineering. When we refer to “scientific,” we indeed aim to invoke the notion of advancing understanding through rigorous, broad-reaching theories rather than focusing on isolated technical challenges. However, we also aim to ensure that these theories are connected to practical advances and, therefore, encourage a *pragmatic* approach to that scientific endeavor. As you suggest, a more clinical approach to engineering might be a good descriptor of our position, and we are grateful for this distinction. We will consider this point as we prepare an updated version of the text.
>
> **Q1 - Opportunities for in ML theory contributions**
>
> Absolutely! As discussed in Section 4, simplified deep learning phenomena provide a tractable setting for applying mathematical theory, which can then contribute to both theoretical and empirical advances. There are a number of examples of works that take this exact approach.
>
> For example, the bias-variance tradeoff provides an explanatory theory with broad, established relevance for overfitting in machine learning. Double descent appeared to challenge a conventional understanding of this theory and called for either an update to the theory or some resolution. In [1], the authors were able to analyze this tension mathematically and rigorously update our understanding of this tradeoff which has practical applications. Similarly, [2] were motivated by double descent to deepen our understanding of ridgeless linear models in the interpolation regime where mathematical theory is both possible and a valuable approach for improving this class of models with practical applications.
>
> These examples reinforce the opportunities provided by deep learning phenomena at the intersection of empiricism and mathematical theory.
>
> We thank this reviewer for their thoughtful review and support for our paper. If there are any remaining questions we would be happy to discuss further.
>
> [1] Adlam, B. and Pennington, J., 2020. Understanding double descent requires a fine-grained bias-variance decomposition. NeurIPS.
>
> [2] Hastie, Trevor, et al. "Surprises in high-dimensional ridgeless least squares interpolation." Annals of statistics (2022)

---

> > ### Comment · Reviewer_X44g · 2025-04-05
> >
> > I appreciate this. My only concern is: Will your answer to my question change the text of the paper itself? It wasn't obvious from what you wrote there.

---

> > > ### Author Response · Authors · 2025-04-05
> > >
> > > Thank you for your response.
> > >
> > > Apologies for not being explicit, yes we intend to make appropriate changes to the paper based on this feedback (in addition to the feedback of the other reviewers).
> > >
> > > Specifically, regarding your comments we intend to:
> > > * Add a clarifying sentence on the representativeness of the three discussed phenomena reflecting our response.
> > > * Revisit each instance of our use of the word scientific and change cases in which a more accurate alternative is appropriate (e.g. "engineering"). Furthermore, we will incorporate a clear definition of this terminology in the introduction such that there is no ambiguity about our intended meaning.
> > > * We will extend our discussion in Sec 4 on the intersection of mathematical theory and empiricism to reflect our discussion.

---

### Official Review · Reviewer_JzCY · 2025-03-16

**Significance:** 2
**Argument Clarity:** 3
**Rating:** 2
**Confidence:** 3

**Questions:**

- You argue that double descent, grokking, and lottery tickets rarely appear in large‐scale real-world experiments or can be “tuned away.” Could you clarify how systematically you have verified this? If new evidence showed these phenomena do surface under certain practical conditions, how would that affect your central position?
- Much of deep‐learning research is curiosity‐driven, especially early on when exploring odd behaviors. Where do you see “pure curiosity” fitting into your pragmatic framework?

**Discussion Potential:**

2

**Paper Summary:**

This paper explores the role of surprising or counterintuitive “deep learning phenomena” (e.g., Double Descent, Grokking, and the Lottery Ticket Hypothesis) in machine learning research. The authors note that many of these phenomena are studied as if they posed pressing practical problems. Yet, in typical large‐scale applications, they either rarely appear or can be “tuned away.” The paper’s central position is that rather than striving to “resolve” or “explain away” each phenomenon in isolation, researchers should treat them as “edge cases” or “stress tests” that refine our more general theories of deep learning.

**Position:**

Yes

**Position In Title:**

Yes

**Related Work:**

3

**Strengths And Weaknesses:**

**Strengths:**

- The paper addresses a meta question that has become more pressing as deep learning phenomena accumulate in the literature: “Why spend so much effort on these edge cases, and how can we reap more benefit from them?”
- By walking through Double Descent, Grokking, and the Lottery Ticket Hypothesis, the authors illustrate in detail how each phenomenon is typically absent from large‐scale practice or is mitigated by standard methods (regularization, tuning, etc.). Meanwhile, the authors also discuss how studying these phenomena nonetheless proved influential in informing new lines of work.

**Weaknesses:**

- While the paper’s final section offers guidelines, these guidelines are somewhat broad—mostly best scientific practices (e.g., documentation, reproducibility, hypothesis testing). Concrete, detailed “recipes” on how to unify phenomenon research with truly generalizable insights might further strengthen the paper’s practical impact.
- The authors assume that most phenomenon‐driven work in deep learning is (and should be) primarily guided by pragmatic, utility‐oriented goals. This may understate the importance of curiosity‐driven research in driving unexpected breakthroughs. They do include a brief mention that not all research is purely pragmatic, but a more explicit treatment of how pure curiosity aligns (or doesn’t) with the paper’s recommendations might better engage a broader audience.
- It is unclear whether the position of this paper would have a significant impact on the research community, given that deep learning research is already largely driven by utility. Moreover, it is often very difficult to predict the utility of a research in advance. Most research ends up not being influential, regardless of whether the original goal is pragmatic or not. Meanwhile, a lot of influential research was not deemed pragmatic initially. Thus, there may be little utility in further prioritizing utility (at least the evidence is not clear).

**Support:**

3

---

> ### Author Rebuttal · Authors · 2025-04-01
>
> We are grateful to this reviewer for their thoughtful feedback and appreciate their constructive suggestions.
>
> **W1 - More concrete guidelines**
>
> We appreciate this suggestion and agree that this would further strengthen the paper's impact. In response, we have added a checklist-style table of questions for researchers to consider while working on deep learning phenomena which we hope will concretely encourage more pragmatic research. [This table is provided at this link](https://imgur.com/a/JENjMsk).
>
> **W2 - A more explicit treatment of how pure curiosity aligns (or doesn’t) with the paper’s recommendations might better engage a broader audience**
>
> Thank you for highlighting this. Our intended point here was to reference research that is *exclusively* curiosity-driven without consideration for real-world relevance. For example, in 1796 Gauss proved that a 17-sided polygon can be constructed with just a compass and a straightedge. Although beautiful, we would not expect this proof to provide practical utility. The vast majority of machine learning research, to whom this piece is addressed, is *not* of this sort (Birhane et al., 2022). We also wish to make explicit that curiosity is entirely compatible with pragmatic research, and being innately interested in any topic of research is generally positive. We do not wish to dissuade researchers from pursuing research that aligns with their personal interests, we only encourage a greater consideration for potential practical impact within this research.
>
> We will rewrite AV(a) on L323 to reflect this clarification (e.g. replacing "entirely driven" w/ "exclusively driven").
>
> **W3 - On the utility of prioritizing utility.**
>
> We appreciate the philosophical nuance of this point. This is exactly the type of discussion this position paper was intended to spark! Let us respond on a practical level.
>
> While we do appreciate that estimating the downstream impact of research is challenging, it is certainly not entirely random. The structure of academic funding (both public and private), although imperfect, is typically predicated on our ability to focus research on areas that are most likely to provide practical impact on particular areas of interest to the funding body. On an individual level, researchers are constantly choosing which research questions to pursue. As with other research outcomes such as "publication standard findings" or "a high citation count", we believe a researcher's ability to predict potential for "greater downstream utility" to be greater than chance and suggest that a *noisy* estimate of utility should not be mistaken for a *random* estimate.
>
> For the specific area of deep learning phenomena, this is particularly relevant. In this position piece, we aimed to make this point *without* criticizing or evaluating specific works, as we felt it to be a more constructive approach. However, our qualitative impression is that there is significant opportunity for greater efficiency in this literature, and these improvements are practically attainable. We also believe this view would resonate with many in the community, with similar sentiments sometimes expressed informally. For example, for grokking, [an author of Kumar et al., 2024](https://reddit.com/comments/1defvmv/-/l8d45rs/) noted that "as a reviewer for ICML, I got to see a truly disproportionate amount of low quality submissions about grokking, [...] it is somewhat of a nexus for quackery." Elsewhere, [a recent blog post](https://lesswrong.com/posts/GpSzShaaf8po4rcmA) assessed several notable grokking papers and concluded that "the extremely simplified domains in which grokking is often studied will lead to biased results that don't generalize to more realistic setups." Note, we have no connection to the authors of these comments.
>
> We believe the question of our capacity to influence downstream utility is an important point of discussion for which we have an optimistic perspective & hope to spark some constructive discussion. We will expand AV(c) on L354 to more comprehensively address this question.
>
> **Q1 - On phenomena not appearing in large‐scale real-world experiments**
>
> Indeed, typically a negative existential claim cannot be proved (a classic example being "the Loch Ness Monster does not exist"). Supporting evidence is provided by the lack of counter-examples found in the existing literature which we have surveyed to the best of our ability.
>
> If reasonable evidence was provided that these phenomena *do* appear in settings that are relevant to practitioners then our position would no longer apply to these phenomena and, thus, the significance of our paper would be reduced. Our paper specifically addresses phenomena that "are not representative of challenges encountered in real-world applications of deep learning". We can currently point to several prominent examples making this topic relevant to many in the research community, if this was not the case then the paper would not hold broad appeal.

---

### Official Review · Reviewer_cynL · 2025-03-17

**Significance:** 4
**Argument Clarity:** 1
**Rating:** 3
**Confidence:** 4

**Questions:**

Is "ubiquity vs utility", as the paper title highlighed, really best capturing the debate here? I feel for many people it's less important if the phenomenon is "ubiquitous"; people care more about if the phenomenon shows up in "advanced" systems. On the other hand, in some sense it is *your* position that relies on some ubiquity of the theory we hopefully could build from deep learning phenomenon study -- with this theoretical ubiquity we could hope for the pragmatic utility in a broader sense or in the long run. Also, it feels a bit misleading to using "utility" to characterize your position, as the opposing position -- i.e. to resolve or understand the phenomena "narrowly" -- is also utility-driven, perhaps more utility-driven than yours. My understanding of your point is that, you think the value of deep learning phenomenon study lies in the *indirect* utility that, with the deeper scientific understanding of a generic theory we possibly build from such study, we may observe pragamtic benefits when developing some other systems, probably in some other areas even, where the same thoery applies to as well.

**Discussion Potential:**

3

**Paper Summary:**

The paper discusses how and why the community should study deep learning phenomena, i.e., unexpected phenomena observed in deep learning experiments such as double descent, grokking, the "lottery ticket" phenomenon, etc. The paper argues that many of such phenomena do not show up in SOTA or product-level systems but only manifest under certain conditions. Consequently, the paper posits that studies of this kind of "deep learning phenomena" should not aim at resolving the issues from engineering perspective for systems at present, but instead, should be motivated by the possible long-term benefit of deeply understanding these phenomena from science perspective, for systems in the future.

The paper further supports their position by arguing against several alternative viewpoints. It's pointed out that a "pure science" perspective to the problem (which does not pursue pragmatic benefit at all but justifies deep learning phenomenon study with intellectual curiosity only) is not aligned with the mainstream goal of AI/ML research. It's also argued that, despite the substantial uncertainty and inaccurary in projecting the utility of specific deep learning phenomenon works, it's still better to insist prioritizing works with more projected utility, compared with not prioritizing deep learning phenomenon works with utility motivation at all (another alternative view). The paper ends with some operational-level suggestions to the field, such as better benchmarking and more discussions on utility projection.

**Position:**

Yes

**Position In Title:**

Yes

**Related Work:**

3

**Strengths And Weaknesses:**

Deep learning phenomenon research aim at scientifically establishing explanatory theory for deep learning. This is a distinct and also relatively new area whose research paradigm is still under shaping. The paper being reviewed contributes interesting comments regarding how the research paradigm should be formed, pointing out issues in current research practice in this area such as lack of systematic thinking, as well as providing nice reflection on how this area is (supposed to be) motivated. The arguments against the several alternative views also make the discussion more comprehensive. In general, I believe this paper could inspire more discussion on the research paradigm problem, which is very important for a young area like deep learning phenomenon study.

My main concern is on the presentation of this paper. I couldn't get what this paper really wants to advocate until I finished reading section 2. I believe the introduction section, the abstract, or even the paper title, deserve some significant revisions; otherwise many readers might give up before they get the main idea of this paper. Even in latter part of the paper, the expressions can be quite confusing from time to time. The below sentence is an example, which is the openning sentence of the paragraph at line 302, and is supposed to give the main thesis of this paragraph, but I failed to understand this sentence at all until I carefully read through the example and all following comments in this paragraph.

> While this distinction might be considered subtle or even obvious, is not reflected in current research practices suggesting an opportunity to refine these practices and increase the value and relevance of these studies. For example ...

**Support:**

3

---

> ### Author Rebuttal · Authors · 2025-04-01
>
> We thank this reviewer for their thorough reading of our paper which we believe provided a fair summary of our work. We also appreciate the constructive feedback providing us with actionable improvements for our exposition.
>
> **Pacing and organization of key points**
>
> Our understanding of the reviewer's concern is largely with the pacing/organization of the key arguments that we make throughout the text. In particular, on first reading, at certain points a reader might find themselves unclear on the intended message with clarifying statements only provided later in the text. While other reviewers didn't raise concerns with clarity explicitly (all scoring "good" for _argument clarity_), we believe it to be a fair point as we shouldn't expect every reader to read at the level of detail of a reviewer. The potential impact of the paper can be improved by ensuring the messaging is as clear as possible to a broad audience -- particularly in the title, abstract, and introduction.
>
> Regarding the specifically mentioned title, we note that the ICML reviewer guidelines require that the title explicitly state the position of the paper. Unfortunately, the position of our paper doesn't perfectly fit into a single sentence without losing some precision. We spent some time considering several possible choices and ultimately settled on this one as we felt it captured our general sentiment without misleading the reader. However, we would always be happy to consider any suggested alternatives. We address the point of _ubiquity vs utility_ later in our rebuttal.
>
> Regarding the remainder of the paper, we are happy to carefully review our current version and make improvements where appropriate. Specifically, we will make the following changes.
>
> * We believe that more tangible examples will help clarify some of the slightly abstract ideas in the paper. Therefore, in the introduction, we will introduce a concrete example of two contrasting research approaches to deep learning phenomena. The positive example aligns with our recommendations and illustrates how double descent has encouraged us to refine our general understanding of the bias-variance tradeoff (see rebuttal to MHTA for more detail). We also provide a negative example, in the form of a "new" unifying theory of grokking and double descent, where we show that both phenomena can be modeled remarkably accurately by tracking the number of prime numbers in the weights of the network. We posit that, despite this theory being technically true, it is clearly not going to produce useful knowledge or methods. [Further detail is provided at this link](https://imgur.com/a/riZpNph).
> * We will expand our responses to alternative views (a) and (c) to fully address common responses that we expect readers might have.
> * We will increase cross-referencing within the text to help navigate the reader (e.g. we will reference an alternative view in the main text where it is most relevant).
> * [We have added a checklist here](https://imgur.com/a/JENjMsk) for self-evaluation while working on these phenomena to ground Sec 4 in concrete questions.
> * We will perform a careful revision of the full text to improve clarity for a more casual reader. Especially aspects of formatting and improving the directness of the text where appropriate.
>
> **Ubiquity vs Utility**
>
> We thank the reviewer for allowing us to clarify this point. Our intention was not to frame the discussion in terms of utility _vs_ ubiquity -- i.e. we don't perceive this as a dichotomy. Rather we suggest that, if practical utility is the goal, *and* these phenomena do not appear in practical settings (i.e. they are not ubiquitous), then they must provide value by means other than their direct resolution.
>
> We completely agree that "people care more about if the phenomenon shows up in *advanced* systems". Our point was exactly that they do not show up -- they are *not* ubiquitous in advanced systems.
>
> Regarding the suggestion that "to resolve or understand the phenomena *narrowly* -- is also utility-driven, perhaps more utility-driven than yours." We are not sure that we agree with the statement as written. Suppose we develop, for example, an accurate metric for measuring double descent using just the model parameters (e.g. the prime number approach from earlier) but (a) double descent itself isn't relevant to advanced systems; and (b) this theory doesn't provide any broader useful knowledge relevant to practical settings. Then we would argue that this narrow effort is *not* utility-driven (where the Cambridge dictionary defines *utility* as "the usefulness of something, especially in a practical way").
>
> Overall, we believe we are largely in agreement on this point with the disagreement only requiring a clarification of semantics. We will ensure the text reflects this clarification and would be happy to discuss further if anything remains unclear.

---

> > ### Comment · Reviewer_cynL · 2025-04-02
> >
> > Thanks for the response. After reading it, as well as the other reviews (and your rebuttals to them), my opinion is kind of reinforced, that this is a paper with good insights and ideas but its presentation may be hindering readers from fully appreciating these ideas, or worse, may lead to significant misunderstandings. Below I'd like to give some more comments for elaboration purpose, and hopefully some of them would be helpful with your paper improvement.
> >
> > First, regarding
> > > other reviewers didn't raise concerns with clarity explicitly (all scoring "good" for argument clarity)
> >
> > Despite their clarity scores, it appears that at least one other reviewer has significantly misunderstood your position, and most of them are challenging about the position of "utility-driven science" which is---to my understanding---not what you truly wanted to advocate at all (instead, you wanted to discuss *what kind of* utility we could or should expect from the particular science about deep learning phenomenon, right?). All of them raised concern on either the support aspect or discussion potential. I rated both aspects ("support" and "discussion potential") as "good", but I see the above concerns in the reviews as direct consequence of the confusing presentation in this paper. I honestly had similar concerns at the beginning, when just finished reading the title, abstract, and introduction part.
> >
> > > Our understanding of the reviewer's concern is largely with the pacing/organization of the key arguments that we make throughout the text ... We believe that more tangible examples will help clarify some of the slightly abstract ideas in the paper
> >
> > Well, examples surely help---personally I got to understand what you want to argue exactly by reading the arguments in section 2 about the three example topics---but I sincerely suggest you to consider re-writing the presentation about the "abstract idea" itself. It is not only about the pace or order of the arguments, but about the arguments themselves.
> >
> > For example, in current abstract you said (a) "*we advocate for a more intentionally pragmatic methodology, prioritizing a focus on practical utility.*", and also said (b) "*they should be studied in a manner similar to natural phenomena where they can act as valuable challenges to our broader intuitions by taking a scientific approach.*" Here, statement (a) sounds like you want to persuade people to move away from the deep learning phenomena you just critized and to focus only on those topics with clear practical utility. Then, statement (b) sounds a bit contradictory to statement (a), to people who believe natural science should not focus on practical utility, at least. While adding examples could indeed help with mitigating misunderstandings, I guess it's also essential, if not more important, to revise the statements themselves which is perhaps the source of those misunderstandings.
> >
> > **Ubiquity vs Utility**
> >
> > >  Our intention was not to frame the discussion in terms of utility vs ubiquity ... Rather we suggest that, if practical utility is the goal, and these phenomena do not appear in practical settings (i.e. they are not ubiquitous), then they must provide value by means other than their direct resolution.
> >
> > Great, if you don't intend to frame the discussion around "ubiquity vs utility", then you probably don't want to characterize your position and the opposing position with these terms (as you did in paper title, for example) because they really make people feel that you *are* framing the discussion that way.
> >
> > Technically, although "*do not appear in practical settings*" is indeed a sufficient condition of "*not ubiquitous*", these two expressions are just conveying different meanings and are not the same. Also, "*means other than direct resolution*" is a much better characterzation of your position, compared with emphasizing the term "*utility*" which is just misleading imho (although it's an incomplete characterization, of course, as it did not say what the "means" is).
> >
> > Overall, I guess the point here is not to clarify your idea for me -- I already got your good idea, after a not-so-pleasant read of the current manuscript, admitedly. It's about how to revise the presentation so that more people can get your idea too, hopefully without a same not-so-pleasant read.

---

> > > ### Author Response · Authors · 2025-04-02
> > >
> > > We are very grateful for this valuable feedback and appreciate this reviewer's deep engagement with our work. We agree with the sentiment that revisions to the presentation will greatly improve readability and will put significant effort into this as we sincerely believe the message in this paper to be important. We will add just a few minor concluding points.
> > >
> > > Firstly, let us provide a *very mild* defense of the level of expected clarity in general (i.e. how it takes some time to fully *get* the key message). While typical ML papers are empirical or theoretical in their communication style, ours required a style closer to what one might find in the humanities. It also presented a relatively complex message attempting to strike a balance between a critique of the current standard and a defense of its potential. In works such as these, it may just be the case that a more careful reading is required to get an accurate sense of the full argument. Again, we *can* and *will* greatly improve the clarity in the submitted draft, but felt that this distinction in expectations might be worth considering in parallel.
> > >
> > > Secondly, some minor technical responses:
> > > * *"you wanted to discuss what kind of utility we could or should expect"* - Actually, we would say both are reasonable responses. Yes, our main point was critiquing the utility being derived from our current research practices in this area. But this required us to at least touch on (Sec 3.1) the meta-question of "what is the purpose of ML research in general?" If the field itself is not motivated by the goal of achieving practical utility as discussed, then there would be little basis for complaining that the subfield of deep learning phenomena is not effectively achieving its potential utility. The fact that some readers might take an interest in this aspect of the paper doesn't seem problematic to us -- this is also why we included AV(a), to explicitly address this response.
> > > * *Statements (a) and (b)* - Indeed, the connection to natural phenomena could suggest this to some readers, which we will revise. Statement (a) was intended as a methodological comment (not pushing for or against the research area's existence) saying that we should be more pragmatic in our general approach (currently many papers tend to have little consideration for why they are addressing a phenomenon at all). Statement (b) answers the question "if they don't need to be resolved, then why study them at all?" with the answer that they can still be used as part of the scientific method to update our broader understanding of aspects of ML that *do* result in practical utility.
> > > * *Ubiquity vs Utility* - We agree and will update the text so they do not appear as a dichotomy.
> > >
> > > We deeply believe in the value of sharing this position with the research community and believe that the ICML position paper track provides a uniquely appropriate venue for that discussion. We are committed to ensuring that the feedback provided in these reviews (particularly concerning clarity) is carefully integrated into the paper to ensure the central message (which was praised by all four reviewers) is communicated as clearly as possible.

---

### Decision · Program_Chairs · 2025-04-26

**Decision:**

Accept (oral)

**Comment:**

The reviewers broadly agreed that this position paper addresses an important meta-question in deep learning research — what is the value (scientific or otherwise) in studying “surprising phenomena,” like double descent and grokking?
The paper's core argument for prioritizing broader theoretical understanding over narrow explanations sparked meaningful discussion among reviewers.
A key strength highlighted by multiple reviewers was the paper's potential to generate productive debate in the community about research priorities and methodology.
The concrete examples analyzing specific phenomena helped ground the arguments.

The main concerns centered on clarity of presentation / potential for misinterpretation.
The authors' rebuttal demonstrated a clear plan to address these issues while maintaining the paper's core message.

Overall, this paper makes a strong contribution to ongoing discussions about the nature of research in deep learning. It addresses a philosophical question which many researchers have considered at some point; it will be good for the community to see this question written down clearly and discussed explicitly.